# Perceptions of Tree Risks and Benefits in a Historically African American Neighborhood

**Alexis Martin [1], Jason Gordon [1], John Schelhas [2,\*] and Tawana Smith Mattox [3]**

1. D.B. Warnell School of Forestry and Natural Resources, University of Georgia, Athens, GA 30602, USA; alexis.martin@uga.edu (A.M.); jason.gordon@uga.edu (J.G.)
2. United State Forest Service, Southern Research Station, Athens, GA 30602, USA
3. Athens Land Trust, Athens, GA 30602, USA; tawana@athenslandtrust.org
* Correspondence: john.schelhas@usda.gov; Tel.: +1-706-559-4260

**Abstract:** An expansive body of research demonstrates the social and ecological benefits of urban forests, although urban tree canopy density tends to be lower than average in areas occupied by marginalized populations. Non-profit organizations and local governments have initiated tree-planting programs; however, some of these programs have encountered local resistance. This study took place in a historically African American neighborhood in the Southern USA with a low tree canopy where residents expressed disinterest in replanting trees following a tree hazard removal campaign led by a local non-profit organization. Employing focus groups and interviews, we explored residents' environmental attitudes and risk perceptions by asking about the risks and benefits of neighborhood trees and barriers to the enjoyment of them. The material and emotional bonds residents have with the neighborhood informed their preferences about trees and green space. Trees were often viewed as hazards and financial risks, although they were an integral part of residents' identities for themselves and their community. The findings suggest that neglecting to look at diverse perceptions will challenge a city's ability to communicate about the urban forest and, therefore, sustainably address disparities in tree benefits and problems.

**Keywords:** urban tree canopy; citizen participation; risk perception; tree benefit; race

## 1. Introduction

Low-income and some racial and ethnic groups often experience disproportionately greater exposure to air pollutants and hazardous land uses, which exacerbate respiratory and other illnesses [1]. Racism, class bias, housing market factors, and high traffic exposure as well as lack of access to health care, good jobs, and grocery stores contribute to this problem. While urban tree canopy benefits might impact disadvantaged populations the most, there is often less canopy compared with other neighborhoods [2,3]. This disparity can be considered an environmental justice issue given the role that urban tree cover can play in health and well-being [4].

In response, several recent studies and initiatives, such as the 2023 Inflation Reduction Act Urban and Community Forestry program, have addressed tree canopy in disadvantaged places. However, residents at times have opposed programs to increase tree cover, suggesting that they view urban trees in complex and nuanced ways [5]. Trees have both advantages and disadvantages (for example, the production of leaf debris) and people in different social and economic contexts will view trees in different ways. Urban forestry programs cannot simply try to rectify measured discrepancies in tree cover by planting more trees in disadvantaged neighborhoods. Community engagement and careful qualitative inquiry that improves our understanding of the ways that residents view trees in their neighborhoods (as influenced by broader contexts) is critical groundwork for urban forestry projects that seek to reduce inequities in urban tree canopy.

## 2. Background

Previous research has demonstrated that the urban forest canopy, and, by extension, forest management of risks and benefits, is often unevenly distributed across the municipal landscape. This research has generally demonstrated a negative relationship between urban tree canopy and areas with socio-economically disadvantaged populations [6–9]. Some studies have suggested that income is more of a determining factor of canopy density in comparison to race or ethnicity [8–14]. Riley and Gardner [15] provided evidence of a negative correlation between urban green cover and percent poverty, racial minority population, education, home age, renter population, and population density. Other research has found tree planting activities tended to decrease as the African American or Hispanic populations increased [14]. This phenomenon was exacerbated in places where canopy cover or income was already low.

Other research has focused specifically on public and right-of-way trees. Some of these studies have suggested lower tree cover on public property in areas with higher proportions of low-income, African American, and renter residents [6,10]. Bruton and Floyd [16] found minority and low-income area public parks had fewer wooded areas and less tree canopy cover. Allegretto et al. [9] found similar results in addition to low-income neighborhood parks tending to be of lower aesthetic quality, having fewer amenities, and providing fewer ecosystem services. In their comprehensive review, Watkins and Gerrish [14] found significant inequities concerning public trees but not private trees,

Heynen et al.'s [17] Milwaukee study revealed that canopy analysis can sometimes be misleading with nuances hidden among the findings. The authors found increased canopy density in African American neighborhoods. However, close examination showed this canopy was dominated by undesirable "volunteer" species on fence lines that established due to a lack of property maintenance. While denser than some other areas of the city, this canopy was of lower quality than the other areas, highlighting the importance of canopy quality (or condition) in addition to density. Importantly, urban minority populations and low-income populations often co-occur [18]. This and other canopy studies suggest low canopy density should be addressed regardless of whether the canopy density is correlated with low income, race, or ethnicity.

While the research on tree canopy distribution is informative and contributes to broad policymaking, understanding of the opinions and needs of the groups in the neighborhoods the studies describe is limited. There has been little work focusing on the sociocultural processes, values, attitudes, behaviors, and social positionality that influence structural inequalities and the ways diverse groups think about urban landscapes [19–21]. Exceptions include Carmichael and McDonough [5], who sought out the historical and cultural context when examining human interactions with the urban forest. They noted that residents demonstrated a rational, experienced-based justification for why they did not want trees during a tree planting campaign led by a non-profit organization. Reasons included a lack of initial engagement in the project, mistrust in local government, and anticipation of city/organization neglect of the tree. Negative past experiences with trees resulting in property damage or physical harm were also cited,

In another example, Rae et al. [22] observed a similar lack of decision-sharing power that led to residents' lack of maintenance of right-of-way trees. Residents were only nominally involved in the program planning resulting in resistance to the campaign driven by territoriality (i.e., identification of the space, which was public, as theirs), aesthetics, place attachment, and responsibility. Qualitative analysis revealed that improved communication could at least partly address residents' reluctance. Such studies ultimately suggest that neglecting to look at diverse perceptions will challenge a city's ability to communicate about the urban forest and, therefore, sustainably address disparities in tree benefits and problems.

This study emerged from a project administered by a non-profit organization to prune and remove high-risk trees in a historically African American (respondents used the terms African American and Black interchangeably. For consistency, we use African American

in the paper, although some quotes use Black.) neighborhood of the Southern United States. Following the removal of the trees, residents told the non-profit organization they did not want replacement trees. Through focus groups and interviews, we observed the context in which residents expressed their attitudes and risk perceptions about trees. Two research questions drove the study: (1) How did residents perceive the risks and benefits of neighborhood trees? (2) What were the barriers preventing residents from fully enjoying the benefits of neighborhood trees? Race was an integral component of the research context, although we did not intend to address a research problem solely concentrated on racial dynamics. Ultimately, this research sought to elucidate ways to better help urban tree managers communicate with diverse communities so they may prioritize equity in the management and distribution of urban tree canopy benefits and risks.

## 3. Materials and Methods

### 3.1. Study Area

This study took place in the West Broad neighborhood of Athens, Georgia, USA. Athens is administered by a consolidated city-county government known as Athens-Clarke County, or ACC. The neighborhood comprised the majority of US Census Tract 9 (Figure 1). West Broad is a historically African American neighborhood with many properties deeded during the post-Civil War reconstruction period. In the 2020 Decennial Census, 74% of the Census tract's population were Black or African American (Table 1). The median household income for this area was USD 17,450 compared with over USD 50,000 in the county as a whole, and 54.1% of the population lived below the poverty line. Twelve percent had a bachelor's degree or higher compared with fifty percent in the county (a large university is located in Athens-Clarke County).

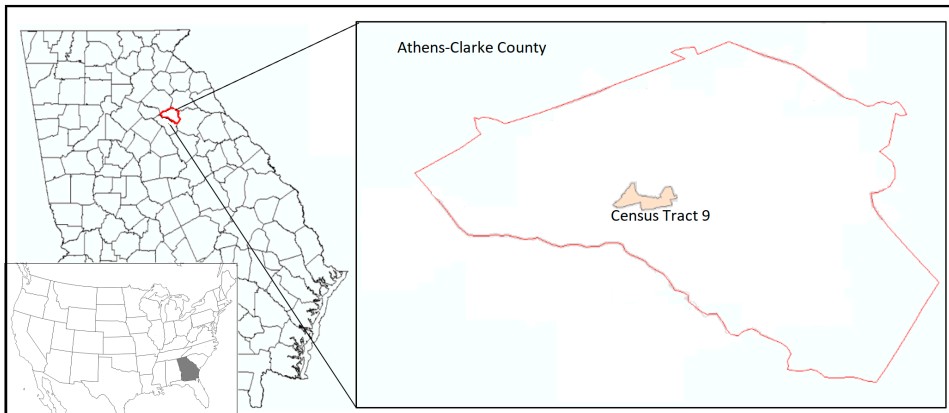

**Figure 1.** Census tract 9 in Athens-Clarke County, GA, USA.

**Table 1.** Demographic comparison of Athens-Clarke County and Census Tract 9 associated with the study site [23].

| Item | Athens-Clarke County | Census Tract 9 (West Broad) | Census Tract Change (%) 2010–2020 |
|---|---|---|---|
| Full-time residents | 126,006 | 4047 | 19 |
| Median household income (USD) | 50,299 | 17,450 | 69 |
| Persons below poverty line (%) | 21 | 54 | −4 |
| Black/African American (%) | 24 | 74 | −6 |
| White, not Hispanic or Latino (%) | 55 | 20 | 6 |
| Bachelor's degree or higher (%) | 50 | 12 | 4 |

A 2021 sample-based urban tree inventory found 58.2% of ACC was characterized by tree canopy cover. Almost two million public trees provided USD 4.3 million in annual tree

benefits [24]. American Forests (2023) [25], a non-governmental non-profit organization, publishes *The Tree Equity Score*, a measurement of equitable tree canopy across a city. The metric is derived from tree canopy cover, climate, demographic, and socioeconomic data with scores ranging from 0 to 100. A score of 100 suggests there are enough trees in the area, low scores (0–69) indicate a high priority for tree planting, and high scores (90–99) suggest lower planting priority. The priority index consists of climate and socioeconomic characteristics with higher priority indicating a more at-risk population based on targets by the US Forest Service and the Nature Conservancy. The West Broad neighborhood had tree equity scores from 73 to 91. This corresponds to canopy coverage between 34 and 47 percent and a high planting priority over most of the neighborhood. People of color, people in poverty, and the health burden index were the primary factors driving the high priority index to increase the canopy cover closer to 50 percent and, therefore, achieve more equitable canopy cover.

### *3.2. Data Collection and Analysis*

We used two qualitative research techniques to explore the research questions: ethnographic photo-elicitation and focus groups which incorporated a mapping exercise. Both methods employed convenience sampling. A critical component of the research was the active participation of a representative (and co-author of this manuscript) of a local non-profit organization which had provided housing, environmental, and food availability services to the neighborhood since 1994. This representative served a crucial role in gaining access to study participants, who may have otherwise been hesitant to speak with outside researchers.

### 3.2.1. Activity 1: Ethnographic Photo-Elicitation

We conducted ethnographic photo-elicitation, a method of interview that uses visual images to elicit comments [26]. This project employed participant-generated photos starting in June 2020 and completed in May 2021 (delayed by almost a year due to the coronavirus pandemic). To initiate data collection, the non-profit representative created a list of adult participants who they contacted to ask for participation in the study. We then distributed one disposable camera with 27 frames to each of 26 participants. There was an equal distribution of male and female participants, and the sample was intentionally restricted to African American residents to focus on their opinions. Participants were between the ages of 18 and 72.

Each participant was given guidelines for taking pictures of "natural" spaces in the neighborhood to elicit their perceived positive and negative aspects of trees and green space in their neighborhood. Guidelines included the following: (1) reserving photos to the neighborhood with a map provided; (2) photos should be meaningful to the photographer; (3) at least one photo in which a tree is the subject of the photo; and (4) take at least 24 photos that show a scene in West Broad. The guidelines helped keep the photos relevant while not influencing the participants through the researcher's viewpoints. Participants had 15 days to take the 27 photos before the cameras were collected. The cameras were then shipped to a private firm for film development, which was returned in the form of digital files.

The photos were then presented to participants in digital form. Using an open-ended interview instrument, each participant explained the context and meaning of their photos. The researcher and participant walked to one or more photo locations in cases where participants were mobile. Interview questions addressed the following: (1) the individual's outdoor experiences in the neighborhood; (2) their perceived meaning of the landscape and associated natural elements like trees; (3) how the neighborhood has changed; (4) the cultural or symbolic meanings of trees; (5) perceptions of tree quality and distribution; and (6) sense of belonging and shared commitment among neighborhood members.

With participant approval, some interviews were recorded to complement the interviewer's field notes, while other conversations were documented via notes taken by the researcher. Discussions were compiled and analyzed for emergent themes using a coding

process involving detailed notes of the interviews, analytic induction, then coding into thematic categories [27]. Using a concurrent and iterative process, we evaluated every new experience articulated by participants based on a short list of tentative codes developed as they emerged from the first five interviews. Ongoing coding was developed through multilevel analysis (analysis was performed simultaneously while collecting data) as the research was being conducted. Themes were compared within and across individuals with key differences and similarities presented in the findings using quotes as illustrations. Ultimately, data saturation (the point at which researchers stop receiving new information) emerged as themes became repetitive between cases. At the conclusion of all interviews, coding among all transcripts was compared to observe for any bias in coding. Relationships among categories were identified and developed into themes, and themes were compared within and across cases. Multiple contacts with the participants helped to validate the information received from the interviews.

### 3.2.2. Activity 2: Focus Groups and Community Mapping

Focus groups helped to validate interview data while allowing new information to emerge. Similar to the interviews, the non-profit representative acted as the liaison by developing the participant list and contacting participants. One session took place in a private home and two sessions occurred in a church commonly employed for community forums. The non-profit representative and one of the authors co-led the focus group discussions. The open-ended question instrument generally reflected the interview instrument. Focus group sessions lasted around two hours and included refreshments. Each session contained 8 to 12 participants for a total of 31.

Focus groups included a one-hour mapping exercise. This exercise allowed for spatial representation of community assets and concerns through discussion. During each session, participants worked collectively using markers to identify key places within a general map depiction of the neighborhood (Figure 2). Participants worked with maps at two stations: one station provided a map eliciting community assets and the second elicited concerns. Facilitators observed participants and took notes of conversations during the exercise. Spatial depictions were compared with the notes. The thematic analysis of notes, recordings, and maps was similar to the process described for the interviews.

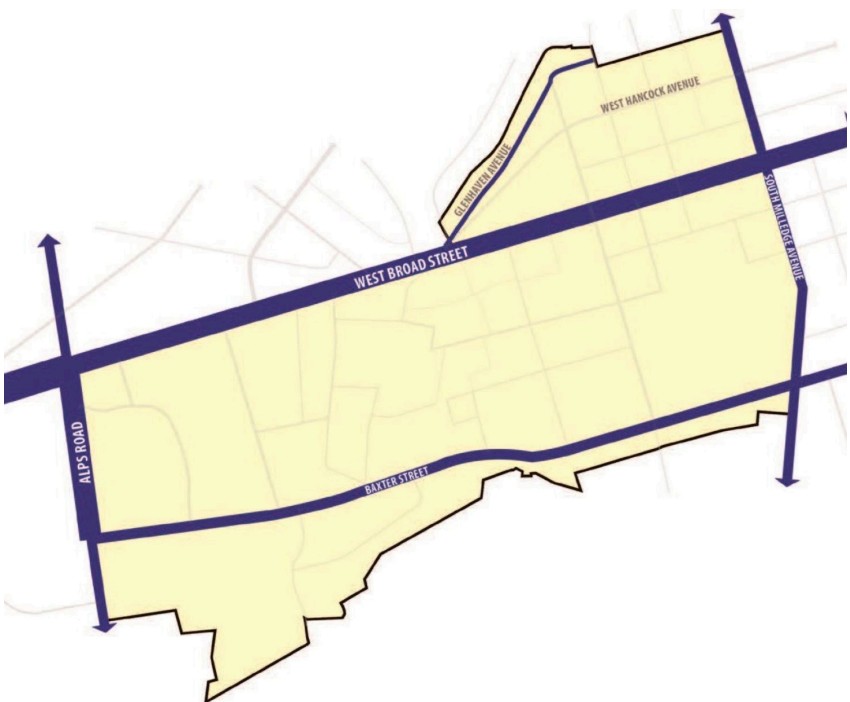

**Figure 2.** Base map for community mapping exercise.

## 4. Results

The results are presented using quotes from interviews and focus groups to illustrate the themes. The primary themes included the following: (1) concerns about trees; (2) competing neighborhood concerns; (3) the functional contribution of trees to the neighborhood; and (4) the contribution of trees to identity. Pseudonyms are used to help protect the anonymity of the participants.

### 4.1. Concerns about Trees

It is important to acknowledge that the participants expressed rational and justifiable concerns about tree or tree part failure, as well as other nuisances such as tree roots in plumbing and raking leaves. For the most part, concerns, and therefore resistance to new tree planting, centered around the factors of age, income, experience with tree failure, and perceptions of the tree's biological integrity. Older participants exhibited increased resistance due to physical vulnerabilities or limitations. For instance, Eva commented, "The backyard . . . it's not maintained really. . . I have to get my grandson to come out and sweep up the pecans because they'll cause you to fall down". Similarly, Mrs. Jones said, "My balance is not as good as it used to be". Tripping on debris and roots was a serious concern for these participants, one of whom lived by herself, and had limited access to quality healthcare. Mrs. Jones went on to describe that stumbling on a root could lead to unwanted medical expenses or immobilization from which recovery would be difficult.

Mr. Walker explained the condition and age of the trees influenced resistance to tree planting in the community:

> The age of trees is pretty consistent throughout the neighborhood. That big oak tree has been here for like 100–105 years. I'm thinking a lot of them are on their way out. . .. That tropical storm came through about 3–4 years ago, one of those limbs got shifted, and that's the only reason it didn't split this house wide open when it fell. To be honest, I don't think residents will want more trees planted. I can't speak for everyone, but I think there's more interest in getting rid of the 100-year-old trees than planting more [due to the age of the tree].

Mr. Walker took a photo of a tree overhanging a house (Figure 3) to show how he was concerned about the high frequency of old water oaks (*Quercus nigra*) and other species in the neighborhood with objectionable risk profiles. Mr. Walker was not unconditionally against tree planting; rather, his quote suggests he was prioritizing mitigation of the risks associated with trees he perceived as dangerous. For this participant, the risks of such trees outweighed benefits such as shade and aesthetics.

By far, one of the major influences on attitudes towards trees was costs associated with maintenance and removal. Economic vulnerability strongly influenced decisions to remove hazardous trees and plant replacements. One participant said this:

> I haven't had [the tree] removed because I can't afford it. . .The roots of this magnolia tree grew into my water line. The councilwoman told me the [root] removal would be about $5000–$6000. I told her, I just won't have any water then. Six grand! I can live off that for the year basically.

This elderly participant demonstrated the critical economic tradeoffs residents faced when addressing their concerns about trees. For many residents, it would be easier and more cost-effective to not have trees at all. Aside from this, the participant notably made a decision based on information he received from an elected official instead of a trained tree worker.

Economic tradeoffs were not only important for fixed income earners. One participant remarked about his daughter and other young people in the neighborhood: "[Living], it takes so much money, young people are trying to provide for their family. They're busy and have to make that sacrifice". A number of participants described how young African Americans from the neighborhood were moving to suburban areas, surrounded by forests, seeking larger lots and homes. Moving to suburban areas is an option, and a tendency,

which perhaps makes it harder for the neighborhood as a whole to focus on trees since older and fixed-income people remain.

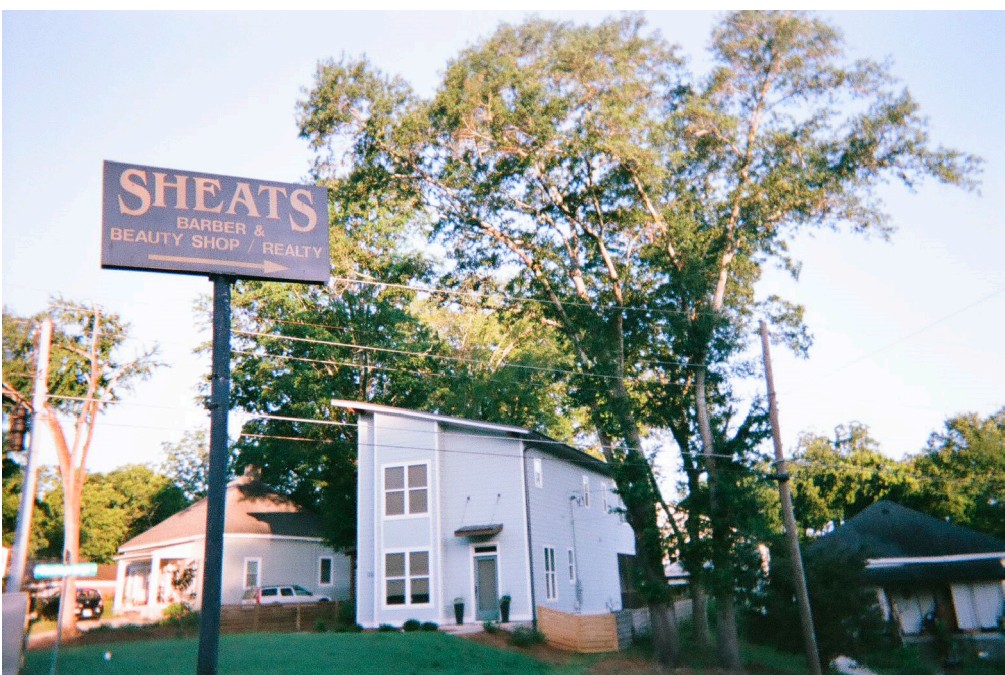

**Figure 3.** Mr. Walker's photo illustrates concern with a tree hanging over a house.

Previous experience with tree failure influenced participants not to plant trees. If a participant had experienced a catastrophic disaster involving a tree, generations of the family or neighbors could express resistance to promoting more trees. Jabari shared a neighborhood story as an example:

> There's a duplex on the corner of Hancock and Chase [Streets]. [Once] there was a real bad storm. It knocked out a good portion off their house and they had to sell it. The tree falling definitely forced them to move. It was a huge house and funds may have prevented them from getting it fixed.

This quote demonstrates displacement as a consequence of tree failure that factored into participants' attitudes about trees. The disastrous loss had both physical and psychological impacts.

Similarly, Mrs. Rosa witnessed a tree failure that temporarily displaced an elderly woman from her home. When asked about tree planting and her perception of trees, she said, "If it's a hazard, that tree needs to be taken out. I don't want a tree to be anywhere that will cause a problem for me or a neighbor". The quote represents a familiar point by participants—particularly elderly participants—that they were not only concerned for their well-being but also for that of their neighbors. In the context of describing the tree failure on her octogenarian neighbor's property, Mrs. Rosa, in her 70s, went on to say:

> As long as Mrs. Smith is here, I won't get out of the neighborhood. I have to help Mrs. Smith. . .I'm not leaving. I am here to make the best place better because it can get better.

### 4.2. Competing Neighborhood Concerns

Our conversations with residents about trees and the natural landscape were nearly always informed by other concerns in their daily lives. Understandably, participants had difficulty separating their thoughts about trees from other neighborhood challenges and opportunities. One participant noted how concerns about crime were intertwined with

vegetation management by referencing a situation involving a respected retired teacher from the neighborhood school:

> I noticed a lot of trees and bushes around Mrs. King's porch. Sometimes she wants to be outside but she's afraid someone may hide behind the bushes– no one should have to live in their home in fear... [The neighbors] cleaned the whole yard, and trees had to go down.

The quote suggests a mental stress accompanying risk perception that could negatively affect one's overall health. In their conversations about trees, crime, adequate health care and education, several participants noted that "it's a lot to worry about".

A major concern was the African American burial site located behind a local government building as depicted in one of the focus group maps (Figure 4). To participants, the cemetery highlighted problems trees can cause. Judith said:

> There's an old cemetery that has been there since before I was born. There was a pest house, a tuberculosis hospital turned military [facility]. Now it's a dead end with a lot of dangerous trees... There's a dead tree, it could've been come down....There are vines growing up the tree, [the vines] choke [the tree] and make them fall...

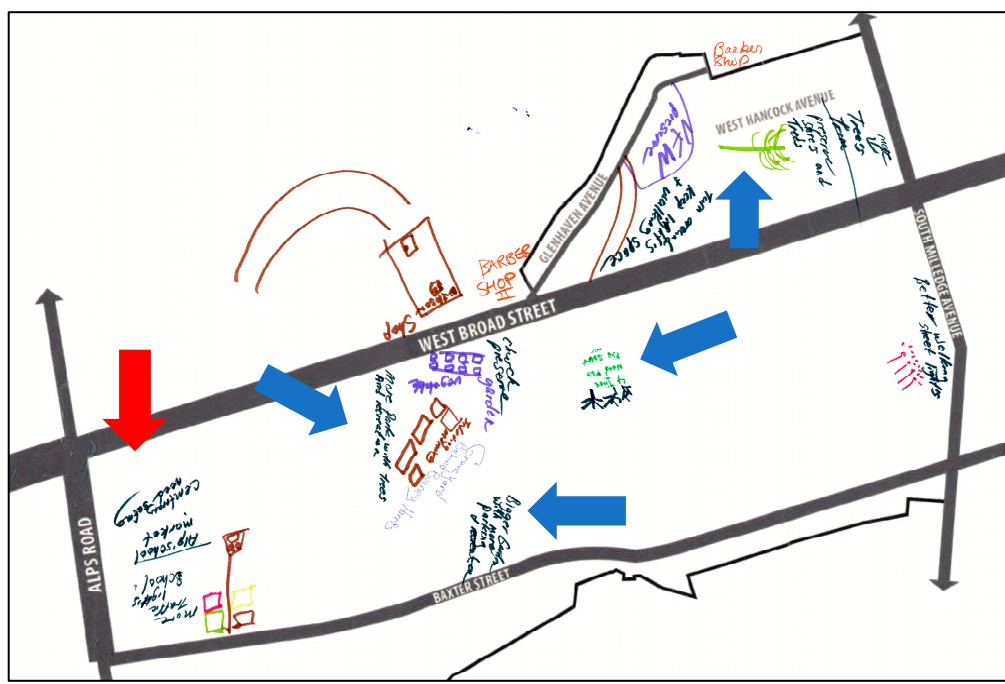

**Figure 4.** Focus group map showing assets and concerns. Trees, probably groves or very mature trees, are noted in several instances (blue arrows). The cemetery is depicted to the left of the image (red arrow), off Alps Road, as a site needing maintenance. Legibility of participants' writing is not necessary for this research article.

During his interview, Mr. King took one of the authors on a tour of the cemetery with the oldest mark dating to 1861. The following quote from Mr. King builds on the condition of the cemetery space and how it affected the entire neighborhood.

> This is a historically Black cemetery, they cleaned it up a lot after [Mrs. Rockman] called, but there's a lot of hazardous wires going through the trees and vines choking the trees. I would like to see it cleaned up more and made more respectfully accessible.

For these participants, the cemetery area has long held symbolic meaning for the neighborhood. The segregated cemetery was located next to a number of "disagreeable"

buildings. Underscoring the lack of consideration for this place where loved ones were buried, it is overrun with vegetation. Concerns about losing the cemetery to overgrowth, a symbol of community history and resilience, were intimately linked to the tree maintenance across the neighborhood.

The overgrowth of the cemetery was a metaphor for the loss of neighborhood identity due to gentrification. The neighborhood had been experiencing a population shift over recent decades as non-Hispanic White homeowners with relatively higher incomes purchased lots. They often renovated the old homes or demolished the old homes to construct new ones. Residents stated the changes might lead to the overall extinction of what once was a thriving culture by severing the thread of long-term interaction and experience passed between generations.

To illustrate, a participant captured the image of a house representing an important communal symbol because a local community leader lived there (Figure 5). According to participants, such symbols will cease to exist, at least as they are currently perceived, as the population of the neighborhood changes. The neighborhood would no longer belong to the people for whom the space was built during a time when, due to institutionalized racism, it was unsafe for African Americans to live anywhere else.

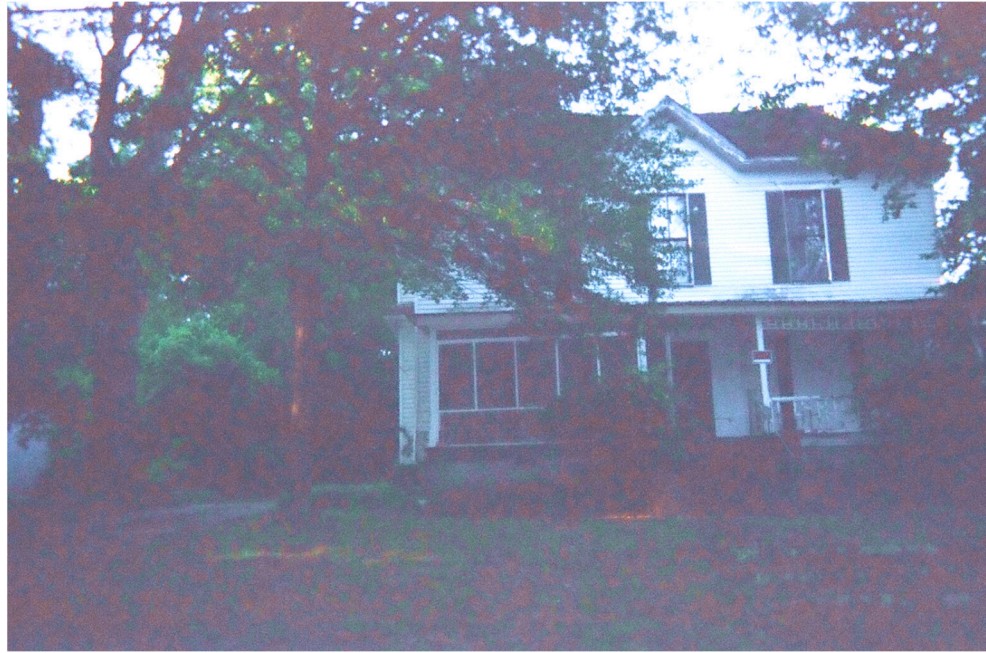

**Figure 5.** Trees in the front yard of a historic home, the former residence of an African American leader.

The next quote by Trenton demonstrates how gentrification led to two conflicting impacts: "The [police] patrol comes through now. And it's because Whites have moved in". Participants described how policing was limited in the past, whereas it improved with the in-migration of an influential social group. Although heightened security was valued, it happened due to gentrification. This contrasts with perceptions that overgrown vegetation, including trees in some cases, was connected with crime. Trenton went on to say, "My dad told me to never let this house get away [the father purchased the house]. I think my daughter will sell it. There's nothing I can do about it". As the neighborhood became safer, and property values increased, younger generations (often non-White) were unable to afford homes. Participants expressed disinterest in tree planting, and otherwise improving the landscape, because they perceived their community supplanted by a new one and their remaining time in the neighborhood as temporary.

### 4.3. The Functional Contribution of Trees to the Neighborhood

Despite their concerns, participants acknowledged the functional benefits of trees such as shade and wildlife habitat. In many cases, they discussed the importance of trees for supplementing dietary needs through the collection of fruit and nuts (particularly southern pecan, *Carya illinoinensis*). Ruby shared her reasoning behind capturing so many pecan trees in her photos, which she stated as being "glorious" (Figure 6).

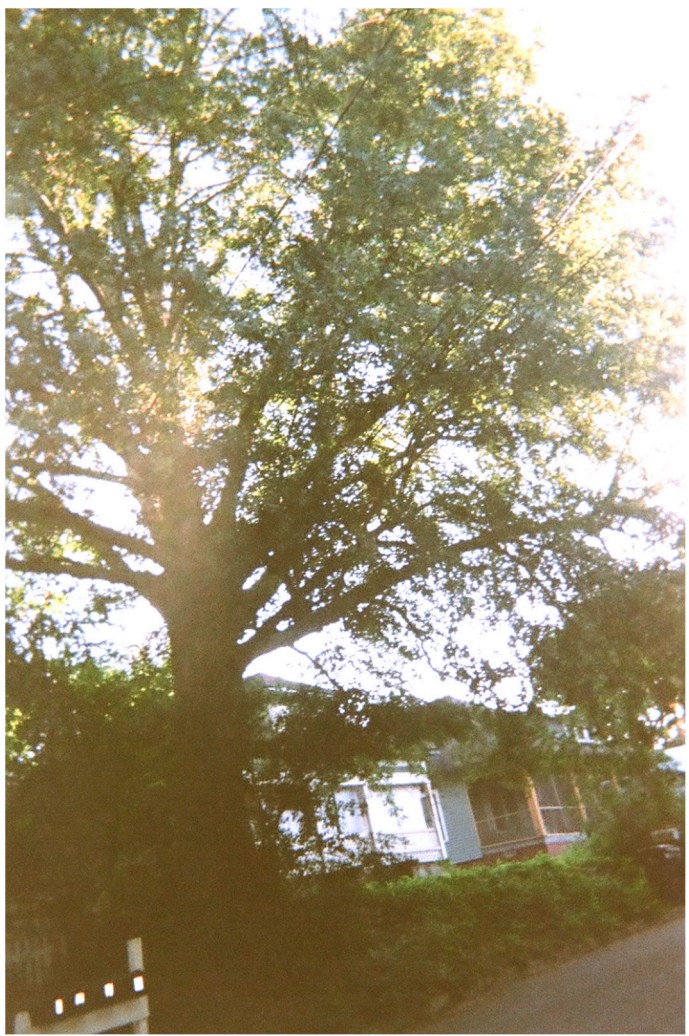

**Figure 6.** This tree was described as a giant pecan and the largest on the street. Pecan trees were associated not only with sustenance but also physical enjoyment of harvesting a natural product.

> Some trees are healthy for the immune system. . . The Native Americans used these native trees for all kinds of things. Medicine. Food. That's another value that as a modern society, we've gotten away from. . .

Her neighbor, Mrs. Rockman, also focused on pecan trees in public spaces, recommending: "Clean some of the smaller trees out and let the big [pecan] trees grow. [So] we'll always have room to do other things with it like a sitting area or something, because without it its hot. . .". Such expressions of value stand in stark contradiction to concerns discussed earlier about pecan trees failing.

Participants also addressed benefits by discussing how trees and other natural elements of the physical landscape served as important components to physical and emotional well-being. Mr. King described the recreation benefits he enjoyed: "I spend about 50% of my time outdoors. I look around and observe. I like to be outside to look at nature and so forth. . .. [Also] for exercise and enjoyment". In the following two quotes, Mr.

Jones references the natural environment as influential in his emotional attachment to the neighborhood.

> I went up to New York with [my wife] for 25 years, but it's not like here. It's not like home. I like it here better. I won't go back... Looking out this front door like I can now. It's just something about it.

> Researcher: What makes it so special?

> [West Broad feels like] a mixture of a warm, homey place, yet still has a city vibe to it. I may be biased, but there seems to be a ray of sunshine that just stays over that area. When the sun shines, other areas don't have that sparkle. It's accessible to so many places [so] it has a city feel [to it] but on the backroads it has a country feel.

After living away, Mr. Jones discovered that the neighborhood was incomparable to any other place and, at least in part, this was due to a country-like character influenced by the landscape. Although the theme was difficult for him to articulate, the quote suggests the neighborhood—in other words, "home"—represented a restorative place that provided a balance between the fast-paced city life and the peace of country living.

A second prominent theme reflecting the functional contribution of trees addressed their role in social interactions and memory. Mr. Walker remembered his membership in a group of neighborhood kids "swinging all the way down to the creek on a tree vine, like Tarzan", while simultaneously learning a lot about nature. This regular experience was an important part of the participant's formative years in which he established friendships lasting a lifetime. Similarly, Jabari would play with other neighborhood kids as they threw walnuts (*Juglans nigra*) at one another, learning in the process that walnut hulls "stain your hand and have a funny smell". Jabari took a photo of a walnut tree to emphasize the importance it had on his development (Figure 7). These stories show how trees contributed to the emotional connections to the neighborhood, despite concerns about tree failure.

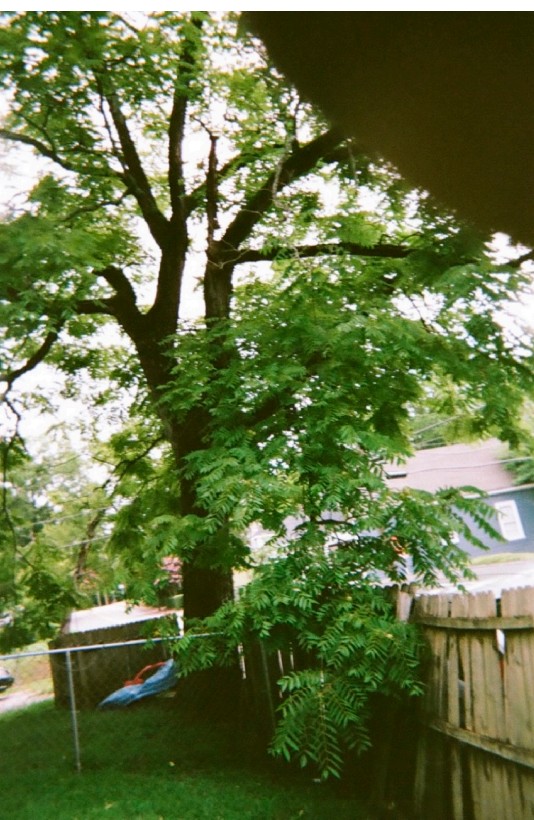

**Figure 7.** Children played under this walnut tree.

### 4.4. The Contribution of Trees to Identity

Despite concerns about gentrification, conversations about the barbershop, the elementary school, or the locally owned general store pointed towards the neighborhood as an important space for the African American community. Discussing one of his photos, Robert said, "You look and see some trees that are pretty beat up. They've been here as long as I have [and have seen some things]". Mr. Gold alluded to difficult times regarding past prejudice as well as people he described as local heroes for combating those wrongs. His photo was important to him because it represented how the neighborhood resisted injustice (Figure 8). The symbolism arose from the perceived age of the tree and its stationary existence. It had lived in the same place in that neighborhood for many decades and "witnessed" society's changes over that time.

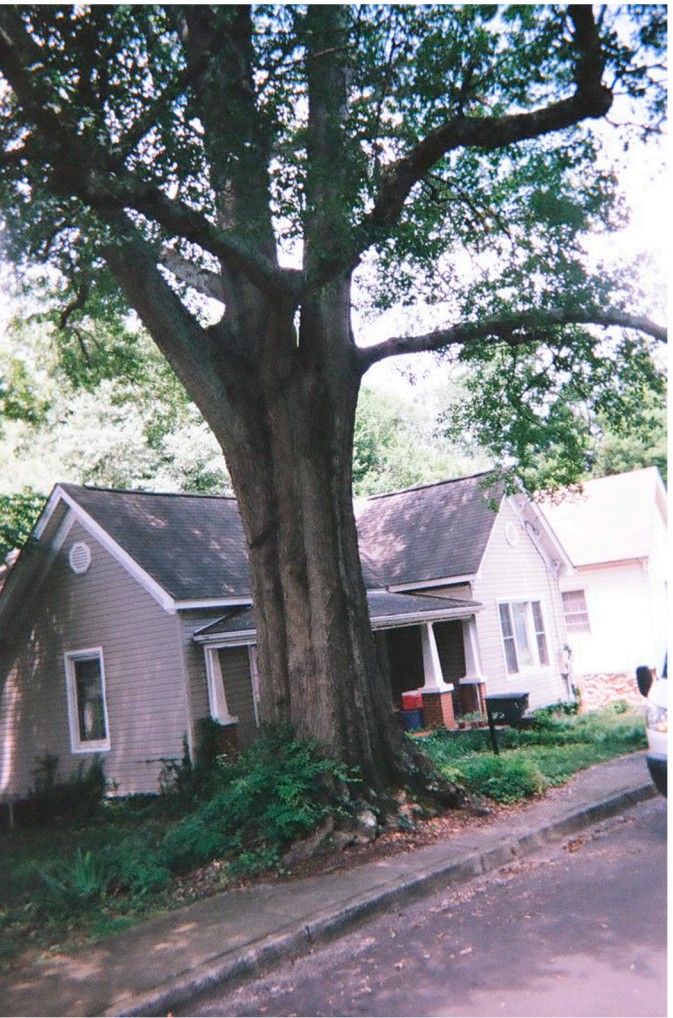

**Figure 8.** This large shade tree symbolized how the neighborhood has resisted racial injustice.

In several instances, the neighborhood was described in terms of individual and collective resourcefulness when public funds were not distributed to African American populations. In the course of discussion, trees were often discussed in reference to other landscape features, particularly gardens. Trees often provided a physical and symbolic frame for gardens, walkways, streams, or parks. One participant noted how her father tended a garden surrounded by trees as part of her family's subsistence: "My dad used to do most of the gardening when he was living. He would get a chair, get in the garden, and we had food".

Likewise, Julia described how her father helped build homes and buildings around the community as he was good with masonry. He was part of the crew who paved

neighborhood roads with rocks. He also grew and sold flowers for the local nursery. Julia proposed that the road at the top of the hill behind the home that her father built should be renamed after her father. When describing what made the neighborhood landscape special, she stated, "that's a part of me—the rocks, the trees".

Most of the participants mentioned some memory, positive or negative, they had about existing or historical trees. As one of the researchers and a participant were walking down a neighborhood street, the participant was thinking aloud: "If trees could talk, they could tell us a lot". He pointed out a large oak across the street from the elementary school that he experienced "young love under".

Such memories connected participants to the landscape with trees as symbols within individual and social memory. Another participant noted one special tree, "There's an oak tree right there, it's been here forever and a day. It's part of the neighborhood [emphasis added]". He might have continued by saying, "it's part of us". Taylor explained his photo of an overgrown wooded area stirred memories of looking for beaver there (Figure 9). Such experiences contributed to pride about the neighborhood and its history. Taylor did not see the area as an abandoned lot with overgrown invasive species as other observers might have. Instead, this was a valued place in which Taylor recalled adventures with family and friends where special bonds formed, and he became a part of that place.

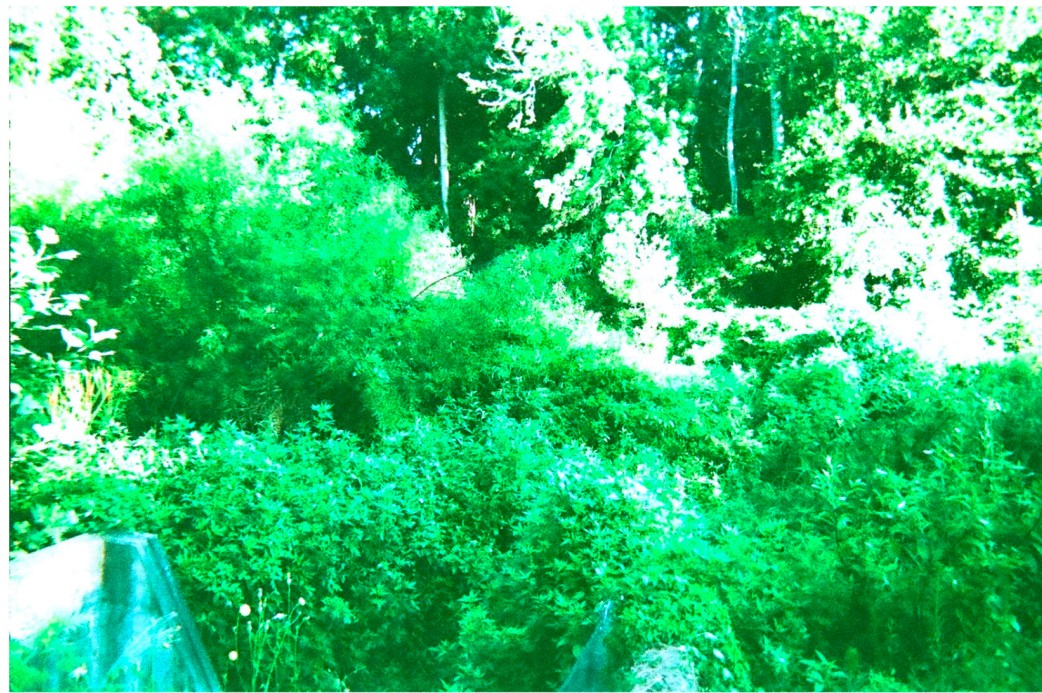

**Figure 9.** One participant described looking for beavers in this overgrown woodlot when he was a teenager.

## 5. Discussion

This study described the context in which residents from a historically African American neighborhood expressed their attitudes and perceptions about trees. There is a large body of work, primarily descriptive in nature, that demonstrates inequitable canopy distribution in areas of low socioeconomic populations [6–9]. By comparison, few studies have focused on the sociocultural processes, values, attitudes, behaviors, and social positionality that influence the ways diverse groups think about urban landscapes (exceptions include [5,19,20]). Carmichael and McDonough [5] examined why under-resourced residents refused to participate in a tree planting campaign despite passionate arguments by city and non-profit organization officials about the benefits of trees and an expanded canopy towards city urban forestry goals. Such arguments are commonplace not only in the United

States but throughout the world. They often draw from a relatively narrow epistemological lens driven by the assumptions of Western science and a highly emotional desire to address a dire environmental problem. Resistance to these well-meaning environmental initiatives, such as tree planting, by under-resourced and minority groups has long been explained by outsiders as a disinterest in trees or, more broadly, a lack of interest in environmental issues [28].

A primary finding of the research is that residents have valid, and sometimes complex, reasoning behind their resistance to increasing canopy cover within their neighborhood, specifically on private property, despite the ecosystem benefits provided by trees (for a review of tree canopy benefits, see [29]). Residents' reservations about tree pruning to reduce risk or replanting after tree removal were not because they did not acknowledge the benefits of trees. Rather, resistance was due to costs associated with tree maintenance, the consequences of tree failure relative to their socioeconomic conditions, and their physical capacity to take care of trees on their property (similar to [5]). In the hope of avoiding tree failure, displacement from their home, or other inconveniences, residents would rather remove declining trees and not plant replacements which they may have to address at a future date.

Attitudes towards trees were not confined to merely two perspectives: being for or against trees. There was an intricate dynamic of factors that influenced resident perspectives that must be considered when planning and implementing urban forestry activities [21]. Our conversations with residents almost always were informed by other concerns in their daily lives. This resulted in discussions about tree canopy reframed to environmental justice rather than solely an environmental or income issue [5]. As such, tree risk perceptions within the community were multidimensional and accounted for other social and environmental risks residents faced in that space. Management strategies and tree planting campaigns that fail to address environmental justice, or at least acknowledge competing local needs, have a lower likelihood of success within under-resourced populations and persons of color. In these neighborhoods, urban forestry must be integrated with other social issues.

Despite concerns, the study participants demonstrated an appreciation for their neighborhood trees based off an emotional bond or connection that individuals developed with the neighborhood's inhabitants and landscape [30]. As illustrated by a number of quotes, this relationship was influenced by experiences, memories, aesthetic and sensory qualities, and interactions with that place over time [31]. The duration of residency and the quality of relationships, as well as the level of personal investment made in shaping or maintaining West Broad as a special place, were also important [30].

Through these strong attachments, participants tended to feel a sense of belonging, identity, and rootedness [32]. Trees, a vehicle for recalling experiences and relationships, influenced how identity was linked to the neighborhood. The seeming constancy of trees, at least their location in the landscape if not in health, contributed to a sense of familiarity when in that place. In turn, the place (and features of the place) held personal significance and symbolic value for participants. Further, participants illustrated how the duration of experiences in the neighborhood, even if interpreted for a time as in the case of Mr. Jones, led to notions of self as a part of the place. Discussions about what trees meant to participants almost always started and returned to events that happened in their youth. In the case of West Broad, trees memorialized respected leaders, ancestors, self-determination, community empowerment, and resilience, particularly during difficult times in the African American experience. Julia's quote about her father exemplified this connection between place and identity.

Although such an influence on individual and collective identity was important to participants' attitudes about trees, we should not overlook the role trees played in how the landscape and its components satisfied functional needs as well [30]. Participants in this study exhibited both physical and psychological needs attributed to the neighborhood space. This dependence was based on specific characteristics of the place which included,

for example, demographic traits of users or the quality of environmental features such as trees and other vegetation. Again, the trees—and their perceived permanency in many instances—framed memories of those who once lived in the neighborhood, motivated learning about and appreciating nature, and provided fruits, ecosystem services, and restorative space to call the neighborhood home.

Participants' identity and functional connections to trees tended to develop largely from positive experiences, which is reflected in the research literature on the connections between places and emotions [32]. However, the demands and stresses of an environment can result in those positive bonds not forming and loss of functional attachment [33]. For example, participants discussed failing trees, leaf and branch debris, and overgrown vegetation that obstructed, or hindered, the formation of positive bonds. Sometimes termed disservices, participants' experiences with these attributes nonetheless contributed to an understanding and connection to the neighborhood and its landscape. We should note that a strong argument exists as to why disservice is an inappropriate term given trees, as non-sentient organisms, do not actionably create hazards or nuisances; rather, these arise only when in conflict with human activities [34,35].

A contribution of this study is the integration of notions about risk perception with sense of place. While concerns influenced by tree condition, age, experience, and income were barriers to fully enjoying trees, residents' risk perceptions and appreciation for trees were mediated by a sense of self as a part of the place as well as the capacity of trees—as a functional component of the space—to satisfy needs (also see [36]). Due to long residency in the neighborhood, participants showed a sense of control. This is a good indicator of their emotional bonds to the place, but also increases familiarity with the sources of risk. Emotional bonds to place may reduce risk perceptions when risk is perceived as less likely [37]. For instance, a sentinel tree that figures prominently in memories about "young love" may be perceived as less of a risk than a tree having no emotional connections and numerous disservices. Conversely, emotional bonds may increase risk perception when the risk is considered highly probable. The results also demonstrated that positive attitudes towards trees were not always easy to articulate due to strong perceptions of risk.

A small body of research has documented associations between place and risk perceptions, typically evidencing a negative relationship [38,39]. In New Orleans, USA, some residents returning home after Hurricane Katrina said displacement was a greater "risk" than threats associated with the storm itself [40]. Negative relationships between place and risk perception may arise because of resource dependence within the space and are highlighted through the ability to withstand hazards with adopted mitigation capacities [41,42]. Regardless of the direction of the relationship, more research is needed to fully understand how emotional bonds to place influence perceptions of risks, such as trees, that residents experience on a daily basis.

## 6. Conclusions

Resident engagement is often discussed as a critical component of community forest management (e.g., [43–45]); however, it is rarely easily accomplished. Sociocultural and economic processes complicate the reductionist interpretation of residents either favoring or not favoring neighborhood trees. Residents have concerns about other conditions (such as crime and education) that can ultimately reduce the positive effect of trees. Simultaneously, emotional community–landscape connections increase benefits. These processes, in turn, reflect cultural and structural aspects of the community which have historical and contemporary foundations. As such, because the results from this study fall into a framework interweaving environmental justice, sense of place, and risk perceptions, they underscore Carmichael and McDonough's [5] argument that residents must be a meaningful part of decisions concerning their tree canopy on public and private landscapes. An acceptable level of collective agreement about natural resource management is particularly challenging during rapid demographic change and population heterogeneity, as was occurring in the study site [46]. Social contexts in which communication is challenged are likely to be best

addressed, firstly, by engaging community volunteer leaders and employing residents in the urban forestry effort.

Social messaging about tree maintenance and planting must address the complex, multifaceted situations within diverse populations, including changing demographics and emotional bonds to place. Such frameworks must encompass equity and how inequity influences the conditions within a community that disrupt sense of place, which, in turn, impacts attitudes about trees. Rethinking the framework of risk perceptions and place to include the dynamics of competing risks and the need for equity among diverse populations is essential to better help urban tree managers communicate with diverse communities so they may prioritize equity in the management and distribution of urban tree canopy benefits and risks.

**Author Contributions:** Conceptualization: J.S. and J.G.; methodology, J.G. and J.S.; formal analysis, A.M. and J.G.; investigation, A.M., J.G., J.S. and T.S.M.; writing—original draft preparation, A.M. and J.G.; writing—review and editing, J.G., J.S. and T.S.M.; supervision, J.G.; project administration, J.G.; funding acquisition, J.G., J.S. and T.S.M. All authors have read and agreed to the published version of the manuscript.

**Funding:** This research was funded by a Tree Research and Education Endowment Fund Research Grant, Project #19-BS-01.

**Institutional Review Board Statement:** The study was conducted in accordance with the Declaration of Helsinki and approved by the Institutional Review Board of University of Georgia Human Subjects Office (VERSION00001582, 23 May 2022).

**Informed Consent Statement:** Informed consent was obtained from all subjects involved in the study.

**Data Availability Statement:** The data presented in this study are available on request from the corresponding author due to privacy reasons.

**Conflicts of Interest:** The authors declare no conflicts of interest. The funders had no role in the design of the study; in the collection, analyses, or interpretation of data; in the writing of the manuscript; or in the decision to publish the results.

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
