# Peer review of "Perceptions of Tree Risks and Benefits in a Historically African American Neighborhood"

_sustainability, doi:10.3390/su16103913_

Round 1

Reviewer 1 Report

Comments and Suggestions for Authors

Dear Authors,

The study has the value of highlighting the need to further investigate the perceptions of residents of a neighborhood on the risks and benefits related to trees and green areas. The study can contribute to identifying what factors make this type of initiative more or less successful.

The investigation was carried out primarily using an anthropological methodology. This is really pertinent. Please think about providing more context for your research focus and hypothesis, such as whether you wish to concentrate on racial component/ethnicity. In fact, the study's findings can help discover more structural concerns that are more focused on other kinds of problems, including a general management strategy, rather than just the racial/ethnic perspective. Where does this kind of stuff, specifically in terms of management, originate? Is it the residents determining the situation? Or their attitude is influenced by this time of structural issues?

Broadly speaking, the racial focus can be controversial and limit the study's robustness in a number of ways. Please, consider to further clarify this and to integrate more considerations from this perspective. The risk is to investigate in the wrong direction. As you know, social groups are very dynamic and many factors and variables must be considered in interaction with this.

Introduction

Line 20: please check for typo (vied)

Line 73-75: This sentence does sound consequently connected with the rest of the paragraph, although it expresses a very relevant point for this analysis.

Lines 78-80: Also, what is relevant is the intention to address the effect and role of structural inequalities as mentioned in the paper “There has been little work focusing on the sociocultural processes, values, attitudes, behaviors, and social positionality that influence the ways diverse groups think about urban landscapes [19, 20, 21].”

Line 135: “People of color”, does really explain the category you are referring to? Please, check for a more consistent way of describing the concept, i.e. your sample.

Materials and Methods

An informative map of the case study area could be helpful. Also, some more demographic data to help highlighting the main changes occurring in this area, as slightly mentioned in some later paragraph of the paper, together with some more geographical information could better help to understand the context.

Line 152-155: The sample is small, but it can already help to highlight some main information. Thus, the description of the sample would be very relevant. In the paper it is mentioned that 26 participants have been involved and that an equal distribution of male and female was ensured. However, could you please explain more about the choice to have only participants from the group that you defined as “African Americans”? You should consider that it might have been interesting to collect information even from residents officially considered as from a different social group living in the case study area. Also, as mentioned, it would be important to explain the social groups aspect: is there a clear stratification in your sample about that?

Although this category (“African Americans”) could be considered as an official one, could this be a kind of distortion in the analysis?

As mentioned in the beginning: what is really your hypothesis? Is there a clear and explicit relationship between the social group, as defined according to other studies, and the perception of green areas?

Have you applied a specific sampling method?

Line 177: How the coding was implemented? Could you include more information about this?

Line 192: You mentioned that in total 31 people participated in the focus group, but your analysed data only for 26 (see line 152-155)? Could you please clarify this point?

Results

Line 204-205: the use of quotes can be relevant but a further systematisation of the data that were collected would have been helpful. How could the results from your analysis be generalised? This would help to make them useful for other similar case study areas.

Lines 350-398: the main focus of this theme is very relevant and in many regards in line with the current debate related to the concept of ecosystem services. For several reasons, your analysis would have benefited from including as much as possible this approach, too. 

I wish you all the best for your research!

Author Response

Reviewer 1

  • Please think about providing more context for your research focus and hypothesis, such as whether you wish to concentrate on racial component/ethnicity.
    • Response: Clarified following the research questions
  • findings can help discover more structural concerns that are more focused on other kinds of problems, including a general management strategy, rather than just the racial/ethnic perspective
    • Response: The revision includes additional verbiage regarding management in in the discussion section
  • Broadly speaking, the racial focus can be controversial and limit the study's robustness in a number of ways. Please, consider to further clarify this and to integrate more considerations from this perspective.
    • Response: We disagree that this paper is controversial, although it brings up important issues in natural resource management that are connected with race. Researchers should not be afraid to bring up these issues. Further, we would ask the reviewer to clarify how addressing race in an Agrican American community limits the robustness of the study.
  • Line 20: please check for typo (vied)
    • Response: done
  • Line 73-75: This sentence does sound consequently connected with the rest of the paragraph, although it expresses a very relevant point for this analysis.
    • Response: Sentence clarified
  • Lines 78-80: Also, what is relevant is the intention to address the effect and role of structural inequalities
    • Response: Sentence clarified
  • Line 135: “People of color”, does really explain the category you are referring to? Please, check for a more consistent way of describing the concept, i.e. your sample.
    • Response: People of color is an accepted term used in North America and elsewhere in the Anglosphere referring to people who are not considered white. The term describes the sample as well as other non-white persons. See Houghton Mifflin Company (2005). The American Heritage Guide to Contemporary Usage and Style. Houghton Mifflin Harcourt. p. 356.
  • An informative map of the case study area could be helpful. Also, some more demographic data to help highlighting the main changes occurring in this area, as slightly mentioned in some later paragraph of the paper, together with some more geographical information could better help to understand the context.
    • Response: Map was added. Demographic changes added to table 1.
  • Line 152-155: could you please explain more about the choice to have only participants from the group that you defined as “African Americans”? Also, as mentioned, it would be important to explain the social groups aspect: is there a clear stratification in your sample about that?
    • Response: Clarification was added in the methods.
  • Although this category (“African Americans”) could be considered as an official one, could this be a kind of distortion in the analysis?
    • Response: We do not understand what the reviewer means by “distortion of the analysis.” African American is the official term used by the United States Census.
  • As mentioned in the beginning: what is really your hypothesis? Is there a clear and explicit relationship between the social group, as defined according to other studies, and the perception of green areas?
    • Response: The Background section lists the two research questions. Hypothesis testing is not appropriate in this form of qualitative research. The literature in the Background section describes associations found in previous studies.
  • Have you applied a specific sampling method?
    • Response: Clarified in the data collection section.
  • Line 177: How the coding was implemented? Could you include more information about this?
    • Response: Clarification added in section 2.2.1
  • Line 192: You mentioned that in total 31 people participated in the focus group, but your analysed data only for 26 (see line 152-155)? Could you please clarify this point?
    • Response: Lines 152-155 in the original file referred to interview participants of the photoelicitation activity. This sample size is indicated in a difference subsection (2.2.1) than the 31 focus group participants the reviewer is referring to (2.2.2).
  • Line 204-205: the use of quotes can be relevant but a further systematisation of the data that were collected would have been helpful. How could the results from your analysis be generalised? This would help to make them useful for other similar case study areas.
    • Response: This qualitative work cannot be generalized, and we did not state generalization as a goal. The goal of this form of research is to explore the nuances of a phenomenon that are may be unobserved in quantitative research. For more information of qualitative research and generalization, the reviewer can refer to Creswell, J. Qualitative inquiry and research design: Choosing among five traditions. Thousand Oaks: Sage Publications, 1998.
  • Lines 350-398: the main focus of this theme is very relevant and in many regards in line with the current debate related to the concept of ecosystem services. For several reasons, your analysis would have benefited from including as much as possible this approach, too. 
    • Response: Additional information added in the Discussion section

Reviewer 2 Report

Comments and Suggestions for Authors

Revision of the manuscript entitled «Perceptions of tree risks and benefits in a historically African-American Neighborhood«

The work developed is a fine presentation of the values and inconveniences of the presence of trees in a neighborhood, according to the perception of its inhabitants.

However, despite the noteworthy content, I do not think that the structure of the manuscript is suitable for the journal. First, there is a complete absence of the numerical information supposedly processed from collected data, apart from the contextualizing information included in the Materials and methods chapter. There are no structured and quantified results that would sustain the discussion. To my mind, the use of interviews with open-ended answers and photographs to initiate comments makes the subjectivity of the analysis and discussion unavoidable. Moreover, the type of language used is more suited to the style of a humanities text.

Therefore, I recommend publishing this manuscript as a book chapter or similar but not as a scientific contribution to Sustainability.

 Finally, I think the following suggestions would be useful to help the authors improve the text:

- line 24: Please use singular form in keywords.

- lines 47-65: This material should be summarized because the main idea is very simple and could be presented in fewer lines and with fewer citations.

- line 108: The heading number is wrong. It should be 3 (not 2) and the subsequent headings should also be corrected.

- line 120: Please do not use uncommon abbreviations in table heading.

- lines 151-152: How were the participants sampled? By subjective sampling? Random sampling? Please clarify.

- line 232: Please use italics for scientific names.

- lines 549-558: Please avoid including citations in the conclusion chapter.

- In addition, I have found some mistakes and error messages (text editor) in the following lines: 20, 327, 328, 439.

Author Response

Reviewer 2

  • there is a complete absence of the numerical information supposedly processed from collected data, apart from the contextualizing information included in the Materials and methods chapter. There are no structured and quantified results that would sustain the discussion. To my mind, the use of interviews with open-ended answers and photographs to initiate comments makes the subjectivity of the analysis and discussion unavoidable.
    • Response: This is a report of qualitative research which is published by this journal. A qualitative approach was appropriate for the research questions and context, including contributing to access to the study group which may have been difficult via a quantitative approach such as a survey. For more information of qualitative research and generalization, the reviewer can refer to Creswell, J. Qualitative inquiry and research design: Choosing among five traditions. Thousand Oaks: Sage Publications, 1998.
  • line 24: Please use singular form in keywords.
    • Response: Done
  • lines 47-65: This material should be summarized because the main idea is very simple and could be presented in fewer lines and with fewer citations.
    • Response: We have decided to keep the text as is because the information is important for the research.
  • line 108: The heading number is wrong. It should be 3 (not 2) and the subsequent headings should also be corrected.
    • Response: Done
  • line 120: Please do not use uncommon abbreviations in table heading.
    • Response: Done
  • lines 151-152: How were the participants sampled? By subjective sampling? Random sampling? Please clarify.
    • Response: Done
  • line 232: Please use italics for scientific names.
    • Response: Done
  • lines 549-558: Please avoid including citations in the conclusion chapter.
    • Response: This is a preference of the reviewer and not unconventional. We have chosen to keep the original text and citations.
  • In addition, I have found some mistakes and error messages (text editor) in the following lines: 20, 327, 328, 439.
    • Response: Done

Reviewer 3 Report

Comments and Suggestions for Authors

Interesting approach and definitely important data since it translates a specific form of perception (communities and people are different and diverse perceptions offer new details and insights to the ongoing process of research).

I do think that the discussion is well articulated with theoretical background presented, but I also would like to see the (more) explored and intertwined with the used references.

What is the main question addressed by the research?

How trees’ canopy influences the perception of risks and/or benefits in a specific Afro-American neighborhood

What parts do you consider original or relevant for the field?

The establishment of a close relation between marginalized areas and trees, accounting the general communal perception about trees’ values, both at a concrete (rational) and even abstract level (emotional)

What does it add to the subject area compared with other published material?

It is a sharped and tuned vision applied to a specific community, but, overall, it shows how particular communal points should be perceived not at the scope of a general lens, since people and communities vary, and have different expectations and perceptions about the world they live and the places they bond with.

What specific improvements should the authors consider regarding the methodology?

There could be some issues about the presented photos, but the truth is they were specifically taken by local inhabitants, to reflect their perceptions/feelings about the thematic.

Please describe how the conclusions are or are not consistent with the evidence and arguments presented.

I do think that the discussion is well articulated with theoretical background, but conclusions could be more explored and intertwined with the used references.

Please also indicate if all main questions posed were addressed and by which specific experiments

Yes, the use of focus groups and interviews allowed the obtention of a general (and personal) perspective about the studied thematic.

Are the references appropriate?

Yes

Comments on the Quality of English Language

Nothing to point out.

Author Response

Reviewer 3

  • I do think that the discussion is well articulated with theoretical background presented, but I also would like to see the (more) explored and intertwined with the used references.
    • Response: Additional discussion and reference to the literature has been included.
  • There could be some issues about the presented photos, but the truth is they were specifically taken by local inhabitants, to reflect their perceptions/feelings about the thematic.
    • Response: Done

Round 2

Reviewer 1 Report

Comments and Suggestions for Authors

Dear Authors,

Thank you for your responses. For sure efforts have been put for developing this study and the paper can contribute to research strand dealing with management of natural resources. As I highlighted in my previous comments, I think that the focus on the racial factor can just marginally help to better address properly the issues related to the management of natural resources. In this regard, I used the adjective controversial. I do agree with you about the fact that researchers should not be afraid of bringing up relevant issues but also, it is always important to choose the right perspective. It can happen that we focus our research on variables that are not really the ones that could explain the studied phenomenon.

Indeed, the perception of people can provide important information but then the challenge is about being able to highlight the actual reasons behind this perception. Is it true that in your study the racial component emerges as the main reason to explain a certain perception that the African-American community has of trees in the neighbourhood? On this regard, more explanation and more evidence would be needed. I think this study needs to be completed and further developed by double-checking what other factors can be behind the perception.

On line 80, you included a reference to "structural inequalities". It would be very relevant to keep this focus in your analysis. In the end, could it be possible to conclude that the results you highlighted in your study, recall factors that are not that much about culture but more structural issues.

All the best for your research.

Author Response

We appreciate the efforts of this reviewer to improve our analysis, but there are several ways in which we felt that the reviewer misunderstands the purpose and content of our paper.

  • This reviewer is skeptical of the role of race in natural resource management. However, there is a large body of literature on this topic that discusses race, ethnicity, and natural resources. Race plays a historic and continuing role through discrimination and resulting structural inequalities, as well as cultural patterns of resource values and use.
  • This reviewer highlights the need to take the “right perspective” to address race and natural resources and suggests that we have not sufficiently examined other factors besides race. After carefully reviewing our paper in light of the reviewer’s comments, we find that we have carefully laid out the purpose of this paper as a qualitative exploration of how disadvantaged neighborhoods, minority and poor in this case, view and experience trees. The logic in the paper is:
    • We cite literature (page 2 and 3, lines 58 to 96) that documents the negative correlation between race and urban tree canopy. We go to some lengths to tease out the complexity and nuances of this relationship in the literature review.
    • We explain the purpose of this study at this study site (page 3, lines 97 to 105). To recap, in this historically African American neighborhood, after removal of dead and dying trees, residents said that they did not want replacement trees. We note on page 4, lines 132 and 133, that trees provide many public benefits, highlighting the importance of this inequity. Other research has found communities declining to be involved in tree canopy improvement projects that were intended to benefit the community and address inequities (e.g., Rae et al. 2010, Carmichael and McDonough2019). Thus, this is a broader question. Reasons for disinterest in tree canopy improvement may be diverse, including lack of participation in program development and failure to use trees that residents value, but the literature suggests that these are often rooted in structural inequality, economics, and racial preferences. This paper is based on a qualitative study to learn how residents of one neighborhood perceived the risks and benefits of trees and the barriers they faced in fully enjoying the benefits of neighborhood trees (lines 99 to 104), with the goal of increasing our understanding of this counter-intuitive finding in the literature and in the history of our study site.
    • This study focuses on a neighborhood that is both poor and African American (lines 112 to 121). This is a common co-occurrence in the U.S. South (as well as urban areas in other regions of the country) due to widespread, historical, race-based institutional and structural discrimination. This is a neighborhood that is receiving economic development and tree canopy improvements from public and private entities largely due to an effort to retain and improve a historical African American neighborhood in Athens, GA. We focused on African American residents, and the African American aspect of the neighborhood does come to the fore in our results, generally in concert with the poverty. Clearly both are important, but they are difficult to disentangle. We discuss various factors in addition to, but also related to, race, including economic disadvantage, structural inequalities, and culture. Because these are interrelated, we do not try to compare them in importance.
    • Our focus is to learn how residents of this neighborhood perceive trees, and explore the nuances—including positive and negative aspects that are important to both the study and practice of urban forestry. Qualitative research, as we have indicated in our earlier response to reviewers, does not permit generalization, but it does provide a more nuanced understanding that is useful to other research projects (including quantitative ones) and natural resource management. The larger lesson from our study is that marginalized or disadvantaged communities—whether due to race, economic status, or other factors—often have distinct perspectives on trees that should be investigated and taken into account.

Regarding structural inequalities: As indicated above, structural inequalities and race are intertwined and discussed throughout the paper. We view them, and believe we have treated them, as interrelated factors in this neighborhood. We also note that in the southeastern U.S., these structural inequalities are rooted in a long history of racial discrimination in all aspects of society, including natural resource management, urban re-development, and provision of services. This history of racial discrimination includes very overt policies such as slavery, Jim Crow laws and policies (institutional racism), unequal educational systems, and redlining neighborhoods to segregate races. These have all produced structural inequalities that have significantly shaped our study site, and we have discussed these throughout the paper.

Reviewer 2 Report

Comments and Suggestions for Authors Unfortunately, the main objections I raised were not taken into account in the revised version.

Author Response

We note that we did make corrections in response to six of this reviewer’s comments. We presume that the “main objection” refers to “the complete absence of numerical information.” We responded that this is a qualitative study, which focuses on words and meanings rather than numbers. We noted that Sustainability has published other qualitative studies, and we cannot make this paper into something that it was never intended to be.

Round 3

Reviewer 1 Report

Comments and Suggestions for Authors

Dear Authors,

Thank you for your exhaustive clarification. One last comment appear relevant to be shared with you. The several studies that you mentioned on lines 58-96 have pointed out what kind of relationship can exist between the economic conditions of neighbourhoods and their access to natural resources. It is agreed that these aspects can already explain most of the differences in people perception about trees benefits or risks. Indeed, as mentioned also in your study, the perception of people towards street trees can be significantly affected by certain type of conditions, such as lower aesthetic quality of green areas, fewer amenities connected to them, and a poorer provision of ecosystem services. Yet, it is definitely not easy to assess to which extend these factors determine the perception. However, the aim of your study was to investigate further the role of socio-cultural aspects. And this is also certainly one factor which is worthy to be investigated. Eventually, according to your results, it emerged that people is not really against trees, but they are more concerned about certain type of conditions that reduce the positive effect of trees (e.g. concerns about crime). Thus, one important conclusion is that people perception can be considered a consequence of these aspects. It can be helpful in the end to explain even more if these aspects can be considered as a reflection of the socio-cultural components (or the racial one). Please, just consider to further explain your opinion about this in the conclusions.

All the best.

Author Response

RESPONSE: We interpret the reviewer’s comment as suggesting that we explain how participants’ concerns about crime and other community issues are socio-cultural influences on their attitudes towards trees or racial (we assume the reviewer uses the word racial in place of structural). In response, we have added the following text in the conclusions:

Resident engagement is often discussed as a critical component of community forest management [e.g., 42, 43, 44]; however, it is rarely easily accomplished. Sociocultural and economic processes complicate the reductionist interpretation of residents being either favoring or not favoring neighborhood trees. Residents have concerns about other conditions (such as crime and education) that can ultimately reduce the positive effect of trees. Simultaneously, emotional community-landscape connections increase benefits. These processes, in turn, reflect cultural and structural aspects of the community which have historical and contemporary foundations. As such, because results from this study fall into a framework interweaving environmental justice, sense of place, and risk perceptions, they underscore Carmichael and McDonough’s [5] argument that residents must be a meaningful part of decisions concerning their tree canopy on public and private landscapes. An acceptable level of collective agreement about natural resource management is particularly challenging during rapid demographic change and population heterogeneity as was occurring in the study site [45]. Social contexts in which communication is challenged is likely to be best addressed, firstly, by engaging community volunteer leaders and employing residents in the urban forestry effort.

Reviewer 2 Report

Comments and Suggestions for Authors

I understand the point of view of the authors, but my opinion remains the same: the main concern about the manuscript is that of the absence of quantitative information, as I assessed in my first report.

Author Response

RESPONSE: The reviewer is insisting that we change the fundamental design for this research. This reviewer clearly does not understand nor value qualitative research. Obviously, this must be executed at the beginning of the research and cannot be done at this stage; therefore, we are unable to satisfy the reviewer’s concern about lack of quantitative data.